# Water pumping in mantle shear zones

Jacques Précigout[1], Cécile Prigent[2], Laurie Palasse[3] & Anthony Pochon[4]

Water plays an important role in geological processes. Providing constraints on what may influence the distribution of aqueous fluids is thus crucial to understanding how water impacts Earth's geodynamics. Here we demonstrate that ductile flow exerts a dynamic control on water-rich fluid circulation in mantle shear zones. Based on amphibole distribution and using dislocation slip-systems as a proxy for syn-tectonic water content in olivine, we highlight fluid accumulation around fine-grained layers dominated by grain-size-sensitive creep. This fluid aggregation correlates with dislocation creep-accommodated strain that localizes in water-rich layers. We also give evidence of cracking induced by fluid pressure where the highest amount of water is expected. These results emphasize long-term fluid pumping attributed to creep cavitation and associated phase nucleation during grain size reduction. Considering the ubiquitous process of grain size reduction during strain localization, our findings shed light on multiple fluid reservoirs in the crust and mantle.

[1] Institut des Sciences de la Terre d'Orléans (ISTO), CNRS-UMR 7327, Université d'Orléans, Campus Géosciences, 1A rue de la Férollerie, 45071 Orléans Cedex 2, France. [2] Univ. Grenoble Alpes, CNRS, ISTerre, F-38000 Grenoble, France. [3] Bruker Nano Analytics, Am Studio 2D, 12489 Berlin, Germany. [4] Géosciences Rennes, UMR 6118, Université de Rennes 1, Campus de Beaulieu, 35042 Rennes, France. Correspondence and requests for materials should be addressed to J.P. (email: jacques.precigout@univ-orleans.fr).

Water is one of the main characteristic components of the Earth. It significantly influences geological processes such as partial melting, ore genesis and the production of earthquakes. Deformation experiments have also shown that water may weaken single crystals and mineral aggregates during ductile flow[1–4], with key implications for plate tectonics. For instance, water/hydrolytic weakening of quartz in the lower crust and of olivine in the uppermost mantle may change the mode of lithospheric deformation from localized (subduction zone, narrow rift) to distributed (ridge plateau, wide rift), or vice versa[5–7]. Although the physical process of this phenomenon remains to be resolved, intra-grain water is thought to enhance the glide or climb of line defects (dislocations) during plastic flow, resulting in mechanical weakening[8,9].

Conversely, rock deformation may have an impact on water distribution. While brittle faults are accepted as controlling the circulation of fluids at shallow depth (< 10 km depth), natural observations—mainly of mid-crustal rocks—suggest that ductile shear zones also drive fluids at greater depth[10]. In the brittle regime, the presence of faults controls the migration of fluids by providing structural conduits or through seismic pumping[11]. But at deeper levels and higher temperatures, these fault-related processes are unlikely owing to plastic flow. At present, two possible mechanisms have been proposed to account for water draining in ductile shear zones, including a 'passive' convergence due to an increase of porosity where grain size is being reduced[12], or 'dynamic' pumping through opening of micro-cavities[13]. Fluid migration is indeed particularly observed where mineral aggregates dominantly deform by grain-size-sensitive (GSS) creep, commonly including a component of grain boundary sliding (GBS)[13–15]. When GBS cannot be fully compensated by diffusive or plastic processes, some micro-cavities open and close in a continuous process because of local dilatancies, giving rise to creep cavitation[16–18]. This might result in fluid pumping—including water—towards the shear zone centre. Nevertheless, only a few vestiges of syn-tectonic water draining have been reported so far, and exclusively for crustal rocks[13,19], allowing doubts to persist about the validity of this latter mechanism.

In this paper, we describe systematic changes in dislocation slip-system and related olivine fabric, combined with enrichments in amphibole, that document syn-tectonic water draining in natural mantle shear zones (Ronda massif, Spain). While water accumulates around ultramylonites that deform by GSS creep, fluid aggregation is coeval with multi-scale strain partitioning between water-poor and water-rich layers during olivine dislocation creep. Providing evidence of long-term water pumping during shear strain localization in upper mantle peridotites, our findings strongly suggest that water converges through the opening of micro-cavities coupled with phase nucleation processes.

## Results

**The Ronda mylonitic complexes.** Located in the internal domain of the Betic Cordillera in southern Spain, the Ronda massif encompasses several kilometre-scale lenses of sub-continental ultramafic bodies (Fig. 1a). During continental extension and subsequent thrusting onto continental crust[20,21], these mantle peridotites were affected by several stages of deformation[21,22] including the development of ductile shear zones in the southwestern massif (Fig. 1a)[23]. In this region, the outcrop-scale features consist of alternating protomylonitic and mylonitic layers within so-called mylonitic complexes. These complexes range from a few centimetres to more than 10 m in thickness and extend up to several hundred meters along strike (Fig. 1b,c). They developed from a protolith of porphyroclastic, spinel-bearing

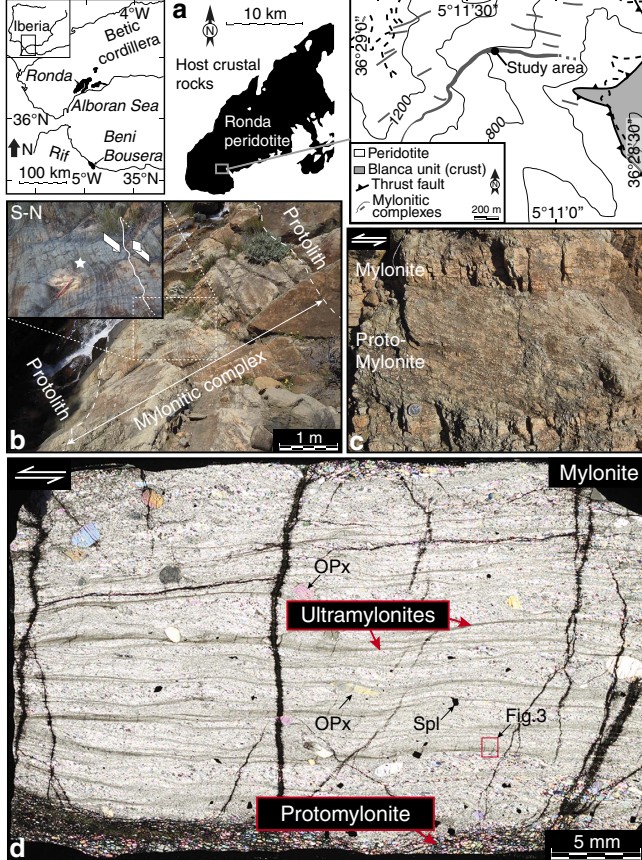

**Figure 1 | The Ronda mantle shear zones.** (**a**) Location of the Ronda peridotite massif (Betic cordillera, southern Spain) and its mylonitic complexes, including the study area (modified from Hidas et al.[23]). (**b**) Studied area of a 5-m-thick mylonitic complex of spinel-bearing harzburgite (protolith) with top-to-the-SW kinematics (inset). (**c**) Detailed view of the strain partitioning between protomylonitic and (ultra)mylonitic spinel-bearing peridotites in the mylonitic complex. (**d**) Photomicrograph (plane polarized light) of a mylonite layer highlighting very fine-grained anastomosing ultramylonite bands adjacent to pyroxene porphyroclasts. The thin section has been cut in the XZ structural plane, that is, the plane normal to the shear plane and parallel to the shear direction. All ductile horizontal structures are crosscut by post-tectonic serpentine. The thin section is 50 μm thick. Opx = orthopyroxene; Spl = spinel.

harzburgite with a foliation oriented 110°/50°NE with top-to-the-SW shear sense criteria (Fig. 1b). The transition from the protolith to a mylonitic complex is progressive and characterized by a thickening of mylonitic and protomylonitic layers (see Supplementary Fig. 1).

In thin section, the mylonitic peridotites contain several ultramylonitic bands of very fine grains that always wrap around pyroxene porphyroclasts (Fig. 1d). Indeed, the protolith and protomylonites have a mean grain size >1 mm for all phases, but the grain size substantially decreases to ~150 μm close to mylonitic layers, ~30–50 μm in mylonitic layers (that is, in between ultramylonites) and <15 μm in ultramylonitic layers, except for a few porphyroclasts of pyroxene (see the Methods section for grain size calculation). At the centre of mylonitic complexes, the density and size of ultramylonitic bands substantially increase, and mylonitic layers contain coarse spinel grains (chromite) with many mode-I cracks sealed by olivine and amphibole (tremolite) (Fig. 2). The occurrence of cracks resulting from tensile stress and sealed by a hydrous phase strongly suggests that the cracking process was the result of high fluid

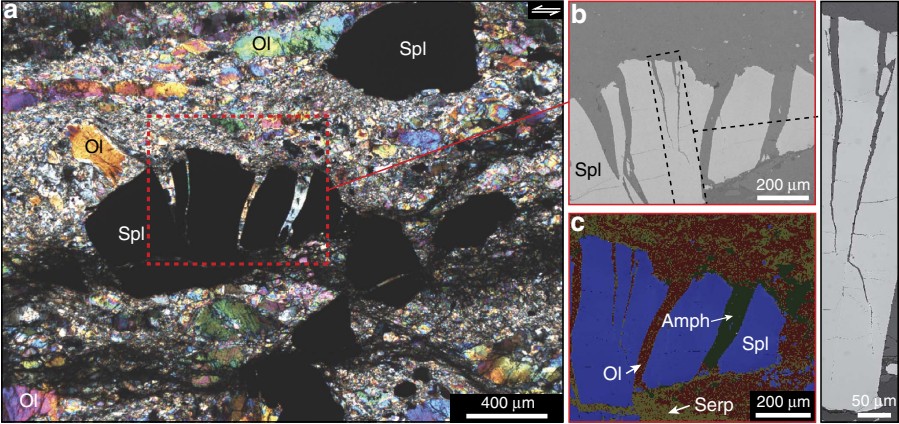

**Figure 2 | Mode-I cracks in spinel located near the centre of a mylonitic complex.** (**a**) Thin section (polarized light) of a mylonite layer containing cracked spinel (Spl; chromite) in between ultramylonite layers (a white star shows the field location in Fig. 1b). (**b**) Backscatter electron (BSE) image showing the mode-I cracks in spinel. See the methods section for analytical conditions. (**c**) Element map of the BSE image using energy-dispersive X-ray spectroscopy (EDX). The colour coding is based on the relative amounts of O, Mg, Si, Fe, Al, Ca and Cr. This map highlights olivine (Ol) and amphibole (Amph) as filling the mode-I cracks. Serp = serpentine.

pressure[24]. Post-tectonic serpentine is also present in the protolith and protomylonites, but is less abundant in mylonites and is quasi-absent in ultramylonites.

Through high-resolution element maps collected by electron microprobe, we further demonstrate enrichment in secondary phases and the presence of amphiboles (pargasite ± tremolite) in fine-grained ultramylonites (Fig. 3a,b). Most peridotites of the shear zones (protolith, protomylonites and mylonites) are indeed composed of olivine (85%) with a small amount of orthopyroxene (13%) ± clinopyroxene (1%) ± spinel (1%), whereas the ultramylonites contain far more orthopyroxene ($\sim$30%) and $\sim$4.5% of calcium-bearing phases. These latter include either clinopyroxene (diopside) or amphiboles with local intergrowths (Fig. 3b). The amount of amphibole generally increases with reducing grain size, but we also document higher content of amphibole in some mylonitic areas located along ultramylonitic layers (see Supplementary Fig. 2). In ultramylonites, all phases are strongly mixed with a low-porosity, with backscatter electron (BSE) images revealing only a few pores of $\sim$1 μm size (Fig. 3c). Very little amphibole is present in the protolith[25].

Based on mineral compositions in the mylonites and ultramylonites and using available geothermobarometers (see Supplementary Tables 1–6), we appraise (1) a pressure reducing from $\sim$1.2 GPa to $\sim$0.5 GPa, and (2) a coeval decrease in temperature from >1,000 °C to below 600 °C with reducing grain size (Fig. 3d). These conditions are consistent with previous estimates that documented decompression and cooling from $\sim$1 GPa/1,000 °C to $\sim$0.5 GPa/<700 °C in the study area[22,25,26]. Using Perple_X calculations to construct a P–T diagram of phase equilibrium (pseudosection), we further give an estimate of the ultramylonites water content. Focusing on a very fine grain size area, we estimated the mineral modal proportions through point counting using element map. This gives 65.9% for olivine, 28.8% for orthopyroxene, 3.0% for amphiboles, 1.5% for clinopyroxene and 0.8% for spinel, which was subsequently converted into oxide percentages. Based on our P–T estimates, as well as calculated isopleths for amphibole and clinopyroxene on the pseudosection, we need $\sim$600 p.p.m. $H_2O$ to reach 3% amphibole (2.25% pargasite + 0.75% tremolite) and <2% clinopyroxene in the ultramylonites (Fig. 3e). As 3% amphibole is close to the maximum that we found in ultramylonites, a lower water content may be expected. However, the absence of

plagioclase necessarily implies a minimum of 400 p.p.m. $H_2O$ (see Supplementary Fig. 3). For further information about microstructures, phase compositions and P–T estimates, the readers may refer to Hidas et al.[25].

**Olivine fabric and lattice misorientations.** Through electron backscatter diffraction (EBSD) analyses, we documented lattice preferred orientation (LPO) and related features for olivine. All LPOs are characterized by olivine [100] axis maxima at low angles to the shear direction (X) and [001] axes close to the pole to the shear plane (Z), which is typical of an E-type fabric (Fig. 4a)[27]. This feature differs from the distal protolith where an A-type fabric has been documented[22,25]. Moreover, in some layers, the [010] and [001] axes are more dispersed and distributed as girdles normal to the shear direction. Together with a strong [100] axis maximum that is close to the shear direction, these features are typical of an axial-[100] fabric or D-type fabric[28,29], giving rise to a combined type of olivine LPO between E- and D-type fabrics.

To better characterize this combined LPO, here referred to as D/E-type fabric (Fig. 4a), we used the eigenvalues of each pole figure to calculate the BA index[30,31]. This index indicates a degree of combination between axial-[100] fabric (BA = 1) and axial-[010] fabric (BA = 0). A point maximum LPO such as an E-type fabric will have an intermediate value, typically around 0.5. Across the Ronda shear zones, the BA index indicates pure E-type fabric at the edge of the protolith along mylonitic complexes (BA = 0.48), and at the edge of the mylonitic layers adjacent to ultramylonites (BA = 0.49) (Fig. 4b). Elsewhere, olivine developed a D/E-type fabric with significant components of D-type fabric, including in the protomylonites (BA = 0.67), the core of the mylonites (BA = 0.73) and the core of the protolith layers located between mylonitic complexes (BA = 0.81) (Fig. 4a,b). We also found D/E-type fabric in the ultramylonites (BA = 0.70), but this value remains uncertain because of their very weak and dispersed LPOs (Fig. 4a).

In relation to this, the strength of olivine fabrics was investigated using the texture[32] (J) and misorientation[33] (M) indices. These both give the degree of mineral alignment from random (J = 1; M = 0) to single crystal-like LPO (J = $\infty$; M = 1). Across the Ronda shear zones, olivine fabrics range from moderately weak LPOs in the protolith (J = 3.87; M = 0.14) to very strong LPOs at the edge of mylonitic areas adjacent to ultramylonites (J = 16.80; M = 0.46). They also systematically

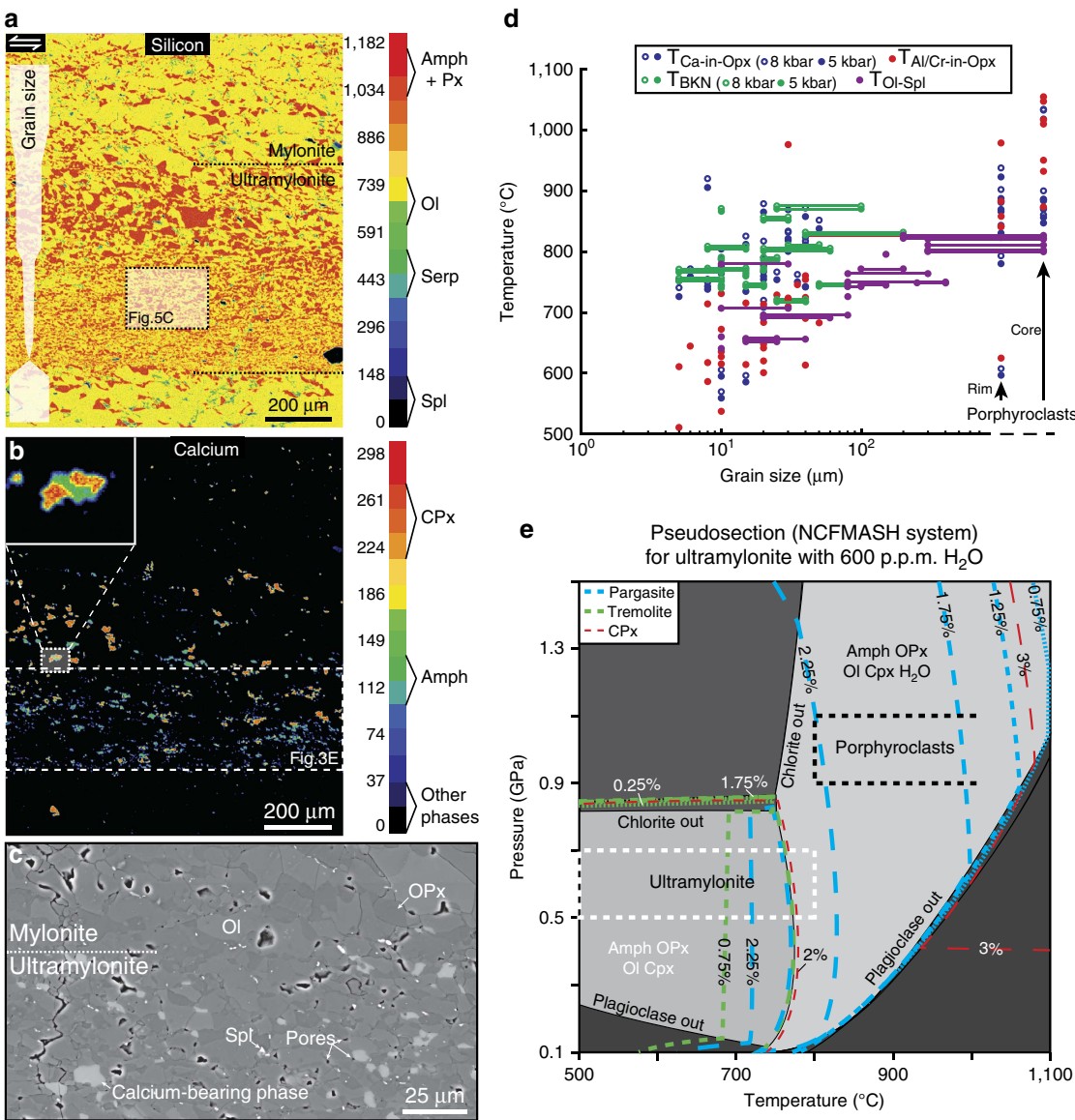

**Figure 3 | Phase enrichment and pressure-temperature-water content estimates.** (**a,b**) Element maps (electron microprobe) that show the distribution of silicon (**a**) and calcium (**b**) across an ultramylonite layer (location in Fig. 1d). The step size is 1 μm and the colour scale refers to the number of counts (see the methods section). These maps highlight enrichment in secondary phases (pyroxene (Px) and amphibole (Amph)) and increasing amphibole (pargasite ± tremolite) content with decreasing grain size. The top inset in **b** gives a detailed view of phase intergrowth between clinopyroxene and amphibole. (**c**) BSE image of an ultramylonite where very few micro-pores (<1 μm) have been identified. The calcium-bearing phases are either amphibole or clinopyroxene grains. (**d**) Calculated temperature in mylonites and ultramylonites as a function of mineral grain size, except for the rim and core of porphyroclasts. For calculations, we applied four geothermometers based on major-element mineral compositions measured by microprobe (see Supplementary Tables 1–6). These geothermometers include the Ca-in-OPx thermometer[65], Al/Cr-in-OPx thermometer[66], two-pyroxenes thermometer[65] (BKN) and olivine-spinel thermometer[67]. The horizontal lines link the two different grain sizes in the case of the two-phase thermometers. For pressure-dependent thermometers, we calculated temperatures at pressures of 5 (empty dot) and 8 (full dot) kbar. Pressure estimates are from Putirka *et al.*[68] (see Supplementary Tables 1–6). (**e**) Equilibrium phase diagram (pseudosection) for an ultramylonite. The pseudosection has been constructed using modal amounts of the phases in the ultramylonite layer shown in **b** (see text and methods section). Based on the P-T conditions recorded in the ultramylonites (dotted white rectangle) and using isopleths of clinopyroxene (dotted red line), pargasite (dotted blue line) and tremolite (dotted green line), we estimate a water content of ~600 p.p.m. during deformation of this ultramylonite band. For comparison, we also show the predicted conditions before deformation based on the P-T data recorded in porphyroclasts (dotted black rectangle). Amph = amphibole; Chl = chlorite; CPx = clinopyroxene; Ol = Olivine; OPx = orthopyroxene; Pl = plagioclase; Serp = serpentine; Spl = spinel.

strengthen from centre to edge within each layer (Fig. 4b). Indeed, the J and M indices never exceed 5 and 0.2, respectively, in the centres of the protolith layers between mylonitic complexes, the protomylonites and the mylonites. In contrast, the fabric strength largely exceeds J = 10 and M = 0.3 at the edge of these layers (Fig. 4a). This distribution (using M index)

correlates with the distribution of shear strain (γ) across the Ronda shear zones, deduced from the angle between the long shape axis of olivine grains and the shear plane (Fig. 4b)[34]. It is also important to note that despite high shear strain in ultramylonites (γ > 5), the olivine LPO remains extremely weak (J = 1.64; M = 0.01).

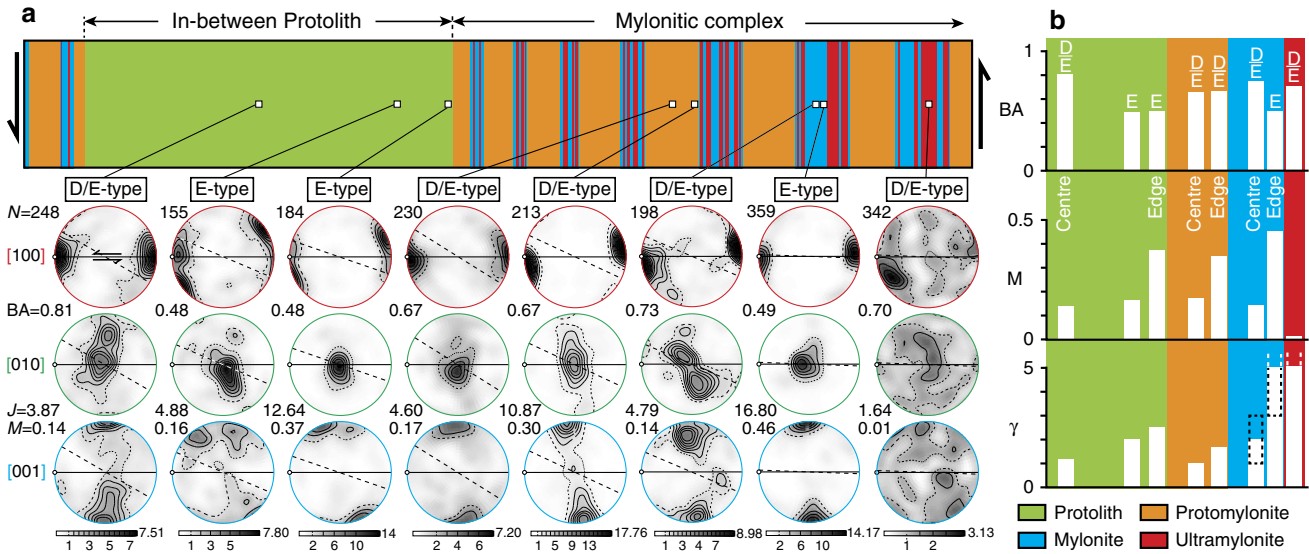

**Figure 4 | Olivine fabric across a mylonitic complex. (a)** Olivine lattice preferred orientation (LPO) across a mylonitic complex, including an in-between protolith. For the protolith, protomylonites and mylonites, we show LPO patterns from centre to edge. LPOs are displayed in equal-area lower-hemisphere pole figures with respect to the shear plane (horizontal line defined by ultramylonitic bands), shear direction (white dot) and foliation plane (dotted line) for the three principal axes of olivine ([100], [010] and [001]). The iso-contours and grey shading are multiples of a uniform distribution based on one point per grain (linear scale). While E-type fabric is identified at the edges of the protolith and mylonites, we observe a combination of E-type and D-type fabrics in other layers[27,28,44]. The BA index gives the degree of combination from $\sim 0.5$ (E-type fabric) to 1 (D-type fabric)[30]. N = number of grains; J = texture index[32]; M = misorientation index[33]; X = shear direction; Z = pole to the shear plane. **(b)** From top to bottom, distribution of the BA index, M index and finite shear strain $(\gamma)$[34] across the protolith and mylonitic complex. We only show the values from centre to edge for each type of layer. In mylonite and ultramylonite layers, the finite strain may strongly vary, including to values higher than 5 (dotted box).

EBSD maps were used to document intra- and inter-granular structural features across the shear zones. These measurements identify olivine sub-grain boundaries in mylonites, with a sub-grain size ($\sim 20\,\mu m$) that far exceeds olivine grain size in the ultramylonites (Fig. 5a). Sub-grain boundaries are also prevalent in both the protolith and protomylonitic bands, in contrast to ultramylonitic layers where only a few sub-grains are observed. The parameters used for the definition of grain boundaries and sub-grain boundaries are available in the methods section.

Of further interest, EBSD maps combined with inverse pole figures (IPF) emphasize different LPOs for olivine as a function of the distance from ultramylonite layers. In Fig. 5a, we display the distribution of olivine crystallographic axes with respect to the Y direction, that is, the deformation axis normal to the shear direction and parallel to the shear plane. As sketched in Fig. 5b, this map highlights different areas that include: (1) ultramylonitic layers with a very weak LPO (J = 2.54; M = 0.04) and a mean grain size of $\sim 12\,\mu m$ ($B_1$ area); (2) mylonitic connected bands adjacent to the ultramylonites with a strong LPO (J = 16.80; M = 0.46) and a grain size of $\sim 34\,\mu m$ ($B_2$ area) and (3) mylonitic areas with a moderate LPO (J = 4.79; M = 0.14) and a grain size of $\sim 33\,\mu m$ ($B_3$ area). The colour coding in Fig. 5a indicates that olivine [010]-axes are parallel to Y in the $B_2$ bands, consistent with the dominant E-type fabric in these areas. This feature is supported by a BA index of 0.49 (Fig. 5b and Supplementary Fig. 4). In contrast, olivine [001]-axes are mostly aligned with Y in $B_3$ areas, giving rise to a different olivine fabric. With a BA index of 0.74 (Fig. 5b), the corresponding LPO is characterized by a combined pattern between E- and D-type fabrics (see Supplementary Fig. 4). Olivine grains in ultramylonites do not show a noticeable LPO, but their LPOs also display a combined pattern of D/E-type fabric with BA = 0.60 (Fig. 5b and Supplementary Fig. 4). Moreover, detailed maps in ultramylonite layers highlight tiny grains of orthopyroxene ($\sim 2\,\mu m$) located

at quadruple junctions of grain boundaries. These quadruple junctions only occur in very fine-grained ultramylonites (Fig. 5c).

For each area ($B_1$, $B_2$ and $B_3$) and the whole map of Fig. 5a, we show the distribution of olivine misorientation axes and related dislocation slip-systems at sub-grain boundaries (Fig. 6). As shown experimentally[35], most sub-grain boundaries are related to tilt boundaries instead of twist boundaries. Plotting misorientation axes on IFPs provides a key to identifying the rotation axis of the main dislocation slip-system[36,37] (Fig. 6a). In Fig. 6b, IPF are displayed for the whole map (in black) and specific areas from $B_1$ to $B_3$, as referred to in Fig. 5b. For the whole map and the B1 (blue) and $B_2$ (purple) areas, the distribution of misorientation axes indicates rotation around the [010] axis, consistent with dominant dislocation slip on the (001)[100] slip-system. On the other hand, misorientation axes are highly concentrated around the [001] axis at intermediate distances from the ultramylonites (red $B_3$ area). This rotation axis is related to activation of the (010)[100] slip-system in olivine[35]. These results reveal, therefore, a change in dislocation slip-system from (001)[100] to (010)[100] in mylonitic layers with increasing distance from ultramylonites.

## Discussion

During mantle flow, olivine aggregates deform by several competing mechanisms that include grain-size-insensitive creep, typically dislocation creep, and GSS creep, such as diffusion creep. GSS creep is also referred to as superplastic flow when GBS significantly contributes to the deformation[15,38]. During grain size reduction, the dominant deformation mechanism commonly switches from dislocation creep to GSS creep, causing a substantial drop in strength[3,4]. Furthermore, intense grain size reduction is often coeval with phase mixing. This provides conditions to keep grain size small by inhibiting grain growth through grain boundary pinning, enhancing GSS creep[39].

In the Ronda shear zones, weak olivine LPO and phase mixing indicate that deformation occurred dominantly by GSS creep in ultramylonites[39,40], and most likely by superplastic flow as suggested by quadruple junctions. These junction types arise when grain boundaries align with the shear plane due to significant GBS[41]. The crystallization of new orthopyroxene grains at quadruple junctions also suggests that phase nucleation occurred during ductile flow. Based on recent experiments[42], it is very likely that strain localization in ultramylonites results from this phase nucleation process and associated grain size reduction. In contrast, strong olivine LPOs and numerous sub-grains indicate that dislocation creep is the dominant deformation mechanism for the rest of the Ronda shear zones (protolith, protomylonites and mylonites). Using the size of sub-grains and an olivine piezometer[43], we estimate a stress between 100 and 200 MPa during deformation.

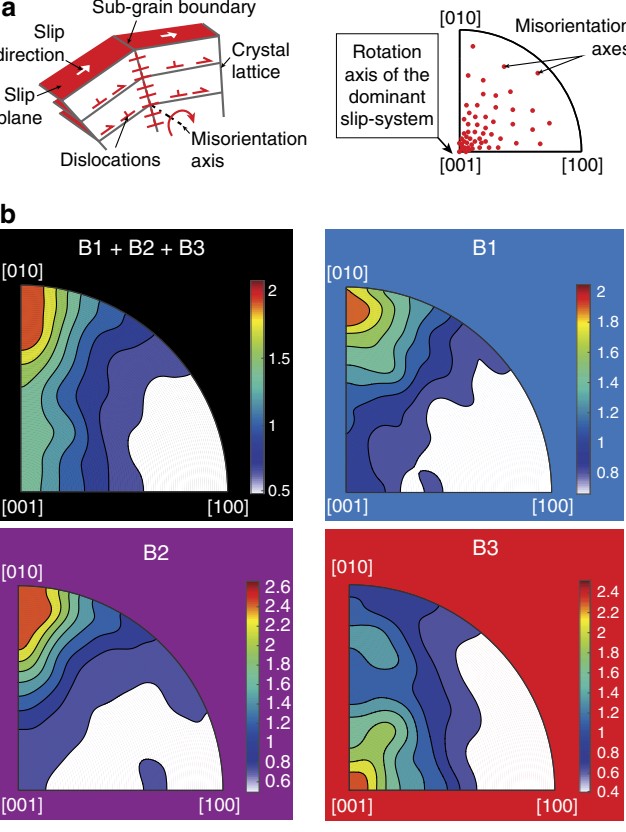

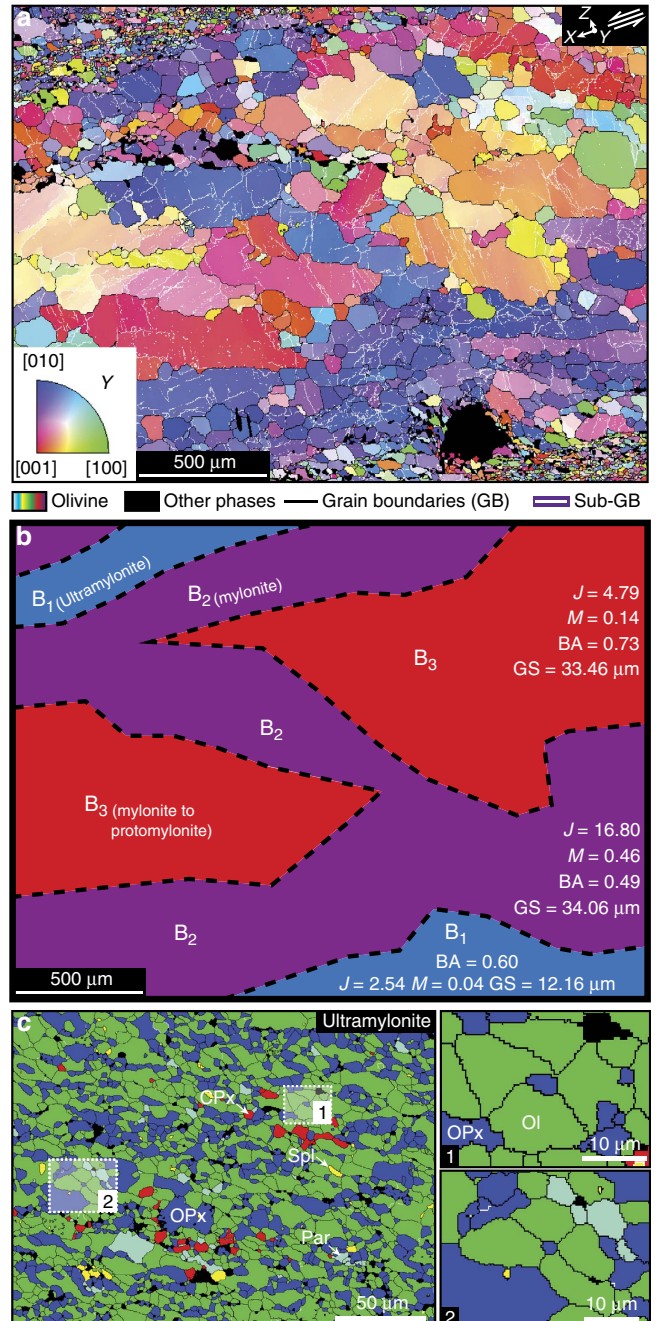

**Figure 6 | Olivine dislocation slip-systems deduced from the distribution of misorientation axes.** (**a**) Sketch illustrating characteristic features of sub-grain tilt boundaries. Inverse pole figures (IPFs) of the olivine axes ([100], [010] and [001]) provide information on the lattice rotation axis induced by the slip of dislocations on a lattice-controlled slip plane and slip direction, that is, on a specific slip-system[35,36]. (**b**) Olivine IPFs showing the distribution of correlated misorientation axes (between 2° and 10°) in olivine for each area illustrated in Fig. 5b, including the whole map ($B_1 + B_2 + B_3$), ultramylonites ($B_1$), purple bands ($B_2$) and isolated areas ($B_3$). The iso-contours and colour scale are multiples of a uniform distribution (linear scale). These IPFs show that isolated areas ($B_3$) differ from the other areas (whole map, $B_1$ and $B_2$) in terms of the dislocation slip-system (see text).

**Figure 5 | Olivine sub-grain boundaries and distribution of lattice misorientation distributions between ultramylonites.** (**a**) EBSD map coloured as a function of the inverse pole figure (bottom left inset) of the [100]-, [010]- and [001]-axes of olivine. The colour coding refers to the orientation of the olivine lattice with respect to the $Y$ axis, that is, the deformation axis normal to the shear direction ($X$) and to the pole of the shear plane ($Z$). The black and white lines highlight grain boundaries (GB) and sub-grain boundaries (sub-GB), respectively (the parameters used to define GB and sub-GB are given in the methods section). Black areas are other phases and non-indexed points (see Supplementary Fig. 4). (**b**) Sketch of the different areas defined by the variation in olivine LPO in **a**, including ultramylonites ($B_1$), adjacent connected bands in the mylonite ($B_2$), and isolated areas between ultramylonites ($B_3$). For each area, we give the texture index ($J$), misorientation index ($M$), BA index (BA) and mean olivine grain size (GS). The corresponding olivine LPOs are available in Supplementary Fig. 4. (**c**) Detailed EBSD map highlighting tiny phases of orthopyroxene (enstatite) surrounded by quadruple junctions of olivine grain boundaries. The location of the EBSD map is shown in Fig. 3a. Black area = pores; CPx = clinopyroxene; Ol = olivine; OPx = orthopyroxene; Par = pargasite; Spl = spinel.

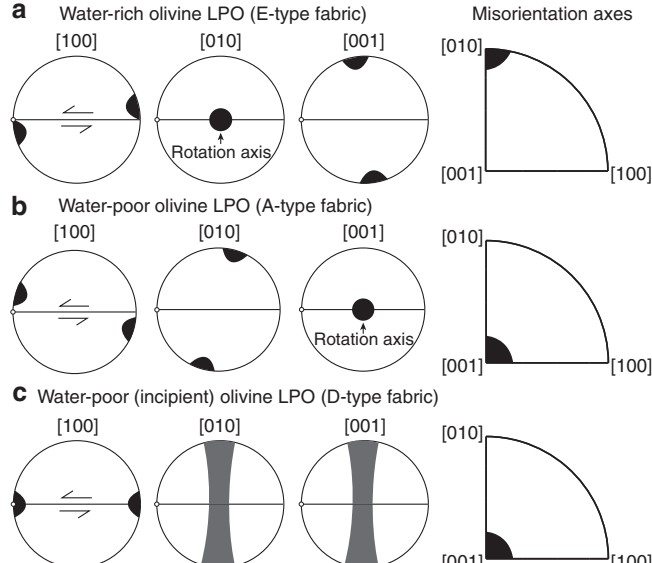

**Figure 7 | Lab-based olivine LPOs and related misorientation axes distribution.** (**a**) Olivine LPO ([100], [010] and [001] axes) produced in water-rich conditions (E-type fabric) with a moderate amount of water[27]. (**b**) Olivine LPO produced in water-poor conditions (A-type fabric)[44]. (**c**) Incipient olivine LPO produced in water-poor conditions (D-type fabric)[29]. While LPOs are displayed in equal-area lower-hemisphere pole figures with respect to the shear plane (horizontal line) and shear direction (white dot), inverse pole figures show the main rotation axis of tilt boundaries for E-type, A-type and D-type fabrics (see text).

When olivine mainly deforms by dislocation creep, several dislocation slip-systems are involved to accommodate plastic strain, but generally, one dominant slip-system governs the development and geometry of the olivine LPO. So far, five olivine fabrics and associated slip-systems have been described (resulting in A-, B-, C-, D- and E-type fabrics). Their preferential activation depends on the deformation conditions including stress, pressure and/or water content[27,44,45]. While E-type fabric is generally recognized as resulting from activation of the (100)[001] slip-system during deformation of olivine containing a moderate amount of water[27] (Fig. 7a), the (010)[100] slip-system is preferentially activated during deformation of dry or poorly hydrated olivine, giving rise to an A-type fabric[44] (Fig. 7b). As for the D-type fabric, it has only been observed for dry or poorly hydrated olivine[28,46] and recent experiments have described it as an incipient A-type fabric resulting from the activation of the same water-poor slip-system[29] (Fig. 7c). All of these fabrics can develop at pressures lower than 3 GPa and below a stress of 300 MPa (refs 27,46).

The development of E-type fabric in Ronda indicates that water was present during strain localization. This is confirmed by the presence of amphibole and our estimate of ∼600 p.p.m. $H_2O$ in the ultramylonites (Fig. 3e). However, the occurrence of combined D/E-type fabric shows that water content has changed during deformation, either from dry to wet conditions (that is, from D- to E-type) or vice versa (that is, from E- to D-type). The occurrence of millimetre-scale water gradients preserved in the mineral fabric (Fig. 5) does not support progressive hydration through water infiltration. Indeed, even in shear zones where very high deformation rates are expected ($10^{-12}$–$10^{-10}$ s$^{-1}$), many years of deformation are needed to generate a mineral fabric, particularly with high fabric strength[29,30]. At a millimetre scale, it is a matter of days to diffuse water through an olivine aggregate, even through the crystal lattice[47]. This rate of water diffusion

excludes *in situ* measurements of water content in our samples (see the methods section). The systematic occurrence of E-type fabric, more or less combined with D-type fabric, also suggests that water was present everywhere in the study area before strain localization, contrary to the distal protolith where a dry A-type fabric has been described[22,25]. The most likely scenario is, therefore, that the D-type fabric overprinted the E-type fabric during strain localization.

The distribution of olivine fabrics implies that water progressively converged towards the ultramylonites during deformation. This gave rise to water accumulation at the edge of the mylonites (E-type fabric), and overprinting of E-type fabric by D-type fabric where olivine was progressively dried (Fig. 8a). Assuming that water mainly diffuses along grain boundaries— grain boundary diffusion is indeed one order of magnitude faster than volume diffusion in olivine[48]—this overprinting is attributed to a switch in the dislocation slip-system where water is being exsolved from olivine, that is, where interstitial water is no longer available (Fig. 8a). We further attribute the D/E-type fabric in ultramylonites to grain size reduction and subsequent water distribution through increasing grain boundary abundance. While water content increases per volume unit, as shown by the distribution of amphibole, it decreases per unit of grain boundary (Fig. 8a). Water convergence is also expected at a larger scale considering the higher density of ultramylonites at the centre of mylonitic complexes. This results in partially dried olivine (D/E-type fabric) in between mylonitic complexes, as well as strain partitioning between water-poor and water-rich layers because of water/hydrolytic weakening (Fig. 8b). The sole exception concerns the ultramylonitic layers, where the expected strain hardening due to olivine drying is far outweighed by grain size reduction and phase mixing, both of which promote weakening by leading to dominant GSS creep[49]. The high amount of water at the centre of complexes further accounts for the occurrence of mode-I cracks in this area, particularly in mylonites where we expect to have the highest amount of water at grain boundaries (Fig. 8b).

Altogether, these features show that water draining in ductile shear zones does not only occur in the crust, but also in the upper mantle, and necessarily on a long-term basis in order to be preserved into the olivine fabric. Our documentation of water accumulation around ultramylonites further excludes grain size reduction as a unique 'passive' process accounting for water draining. They show instead that water converged as a result of 'dynamic' pumping during ductile flow. Although we have no direct evidence of strain-induced cavities, the convergence of water towards layers dominated by superplastic flow supports creep cavitation as the source responsible for water-rich fluid pumping (Fig. 8a). GBS-induced quadruple junctions are also typical sites for cavitation[41]. In addition, the presence of nucleated phases at quadruple junctions, as well as a low porosity, suggests that opening cavities are filled by new precipitating phases, which may enhance long-term water pumping. While creep cavitation is a transient process that first opens cavities, by pumping fluid, and then closes cavities by expelling fluid too[13], the filling of opening cavities by new solid phases precludes fluid expulsion. Lateral fluid-assisted diffusion of mineral components is likewise expected to compensate for the local deficit of fluid solute induced by phase nucleation[25,50]. This accounts for enrichment in secondary phases within ultramylonites, particularly for orthopyroxene, which only relies on increasing silica content.

To conclude, we here ascribe dynamic long-term fluid pumping during the deformation process of GBS and related creep cavitation, both arising from grain size reduction through phase nucleation. As grain size reduction is a ubiquitous feature

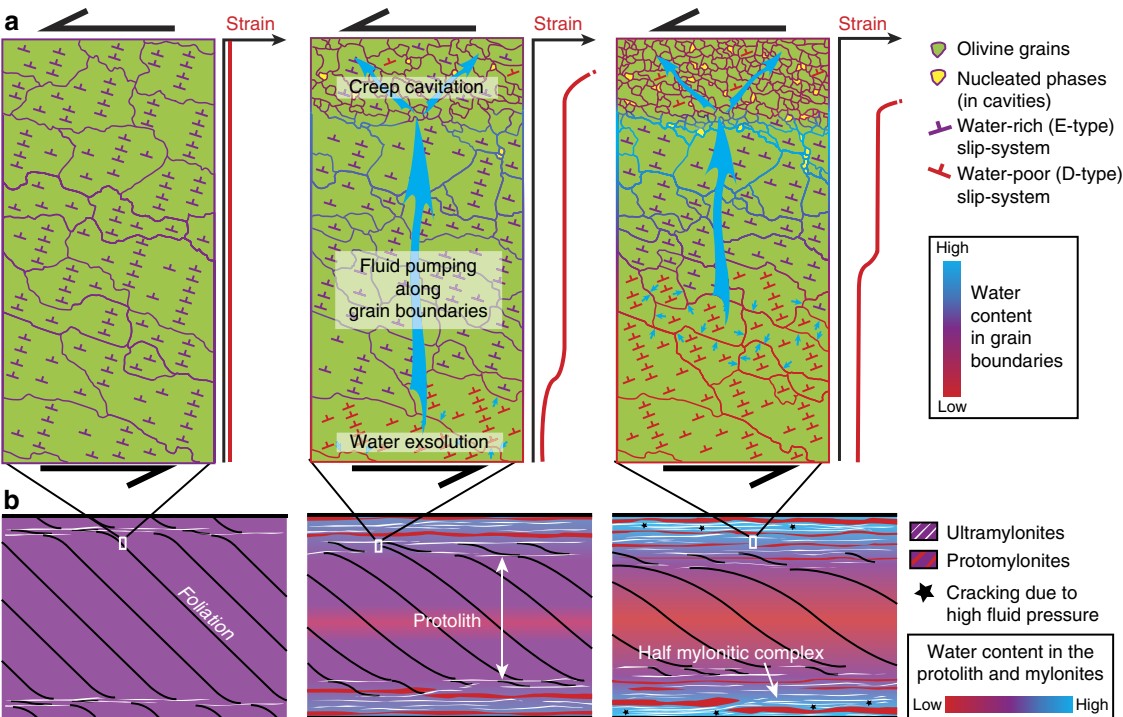

**Figure 8 | Olivine fabric and strain partitioning resulting from water pumping in mantle shear zones.** (**a**) Water distribution in grain boundaries during strain localization, grain size reduction and related water pumping (top part). While water equally distributes on olivine grain boundaries during incipient strain localization, giving rise to an overall E-type fabric, the fluid subsequently accumulates in ultramylonites because of 'dynamic' pumping. The blue arrows refer to water migration through grain boundaries. Accordingly, where water is no longer available at grain boundaries, olivine grains are progressively dried out through water exsolution, leading to strain partitioning (middle and right panels) and E-type fabric that is then overprinted by D-type fabric (change in dislocation slip-system). The D-type fabric also arises in ultramylonites as fluid distributes in new grain boundaries. We here attribute water pumping to creep cavitation and related phase nucleation in layers dominated by grain-size-sensitive creep (ultramylonites). (**b**) Large-scale fluid pumping and strain partitioning in mylonitic complexes. From left to right, water converges towards the centre of the mylonitic complexes, which progressively widen through lateral production of further ultramylonite layers. As a consequence, the olivine fabric switches from an E-type to a D-type LPO in the centre of the protolith and water accumulates in the mylonites, where the density of ultramylonites increases. This gives rise to cracking (black stars) due to high fluid pressure in these zones.

of mantle and crustal shear zones[39,40,51,52], this process is expected to occur wherever strain highly localizes, such as plate boundaries and at the base of the lithosphere-asthenosphere boundary. Our study therefore confirms a ductile process that could provide important fluid reservoirs, impacting the water distribution and related features of both the crust and mantle.

## Methods

**Scanning electron microscope.** BSE images were acquired on polished (using 1 μm diamond paste) and carbon coated (20 nm) thin sections using a TESCAN scanning electron microscope (SEM) at ISTO/BRGM (University of Orléans, France). The SEM was set at an accelerating voltage of 20 kV, a probe current of ~6 nA and a working distance of 10 mm. We also used the TESCAN SEM at ISTO/BRGM with an accelerating voltage of 15 kV to perform energy-dispersive X-ray spectroscopy (EDS) on cracked spinel.

**Microprobe.** Major-element compositions and element maps were acquired using a CAMECA SX-five electron microprobe at ISTO/BRGM (University of Orléans, France). The analyses were performed on carbon-coated (20 nm), polished (using 1 μm diamond paste) thin sections using an accelerating voltage of 15 kV, a probe current of 10 nA and a beam diameter of 1 μm. For point analyses, we applied a counting time of 10 and 5 s for peak and background signals, respectively. Calibration was carried out against natural and synthetic minerals, including albite, $MnTiO_3$, $Fe_2O_3$, MgO, $Al_2O_3$, andradite, orthose, $Cr_2O_3$, NiO and vanadinite.

**Perple_X calculations.** The pseudosections were calculated using Perple_X (ref. 53) with the thermodynamic database of Holland and Powell[54]. It was built in the NCFMASH ($Na_2O$-CaO-FeO-MgO-$Al_2O_3$-$SiO_2$-$H_2O$) chemical system. The weight percent of oxides was calculated by combining the volume percent of mineral modes, which were determined by point counting on elements maps,

and the composition of mineral phases determined at the microprobe. No phase was excluded, but we did not consider chromium in spinel for simplicity. The solution models are from Holland and Powell[54,55] and Holland et al.[56], except for antigorite[57], amphibole[58] and plagioclase[59].

**Electron backscatter diffraction.** The EBSD data were collected using an SEM either coupled to an EDAX Pegasus system at ISTO/BRGM, or using an e⁻ Flash[HR] detector at Bruker Nano Analytics GmbH (Berlin, Germany). Crystallographic information was also combined with EDX (Energy-Dispersive X-ray) spectroscopy to identify phases. The EBSD measurements were performed at a working distance of 15 or 20 mm with an accelerating voltage of 20 kV and a probe current of ~6 nA on polished thin sections (diamond paste of 0.25 μm followed by colloidal silica). In order to avoid indexing errors, we constructed pole figures using one measurement per grain collected manually. The iso-contours and grey shadings on pole figures were plotted using the Unicef careware software package written by David Mainprice (www.gm.univ-montp2.fr/PERSO/mainprice/W_data/CareWare_Unicef_Programs). We further used the open-source matlab-based MTEX toolbox to construct IPF of misorientation axes and to calculate the texture (J) and misorientation (M) indices[30,60]. To plot pole figures and calculate the J index, we used a Gaussian half-width angle of 10°.

For acquisition data processing and analysis of EBSD maps, we used the OIM Analysis 7 software (EDAX) and/or HKL Tango channel 5 system (Oxford Instruments). The indexing hit rates were higher than 80% for all maps, including cavities. We used a minimum of six indexed bands and a maximum band mismatch of 2°. The outlier pixels and zero solutions were removed up to a maximum pixel count of 10. Grain detection was performed using correlated misorientation angles higher than 10° for grain boundaries and between 2 and 10° for sub-grain boundaries. Each grain contains a minimum of five pixels considering a hexagonal grid. The average grain size corresponds to the mean equivalent diameter calculated using the grain ellipsoid fit. EBSD maps were used only to calculate the grain size in the protomylonites (edge), mylonites and ultramylonites. In the protolith and centre of protomylonites, we applied the length-intercept method on BSE images because of their larger grain sizes.

**In situ water content.** A major problem with *in situ* measurements of water content in natural rocks is how far the measured value reflects the water content at lithosphere depths. It has been demonstrated that for orogenic peridotites where the exhumation is slow compared to xenoliths, there are limits in the use of Fourier transform infrared spectroscopy—or other analytical methods—in the determination of olivine water content[61,62]. Water from crustal fluids may be indeed introduced or may precipitate as fluid inclusions and hydrated phases in olivine during exhumation[63]. Note that the Ronda peridotites have been lying on the crust of the internal Betics since more than 20 Ma (ref. 21). Furthermore, lab experiments have demonstrated that water diffusion is fast in olivine ($10^{-10}$ $m^2$ $s^{-1}$) relative to geological strain rates[47]. If water was originally present in olivine grains, it could have been easily diffused out during the peridotite exhumation stage[64]. The presence of post-tectonic serpentine in the studied peridotites may also have strongly modified olivine water content after deformation. It is therefore very likely that *in situ* measurements in olivine of the Ronda peridotites would not reflect the water content of the minerals prior to exhumation, and particularly during their deformation. In contrast, lab experiments have shown that olivine LPOs are sensitive to water content[27,44]. As they directly arise from rock deformation, they provide a more robust method—yet qualitative—to document syn-tectonic water content in upper mantle rocks[61].

**Data availability.** The authors declare that most of the data supporting the findings of this study are available within the paper (and its Supplementary Information files). The source files of EBSD data are available from the corresponding author on reasonable request.

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

## Acknowledgements

We thank Carlos J. Garrido for his help in acquiring field data. We also thank Holger Stünitz, Nicole Le Breton, Alexandra R. L. Kushnir and Jessica M. Warren for their help and comments, as well as Florian Fusseis for his fruitful reviews. This project has been funded by the ETH Zürich (ETH fellowship application FEL01 09-3), ERC RHEOLITH (Grant n° 290864), Labex Voltaire (ANR-10-LABX-100-01) and ANR DSP-Tibet (ANR-13-BS06-0012).

## Author contributions

J.P., C.P. and A.P. contributed to acquire field data. J.P., C.P. and L.P. contributed to performed EBSD analyses. J.P. and C.P. contributed to performed microprobe analyses. C.P. contributed to perform pseudo-sections with Perplex. All authors contributed to the conception of the study and writing of the manuscript.

## Additional information

**Competing interests:** The authors declare no competing financial interests.

