## [Peer Review File · Nature Communications]

Reviewers' comments:

Reviewer #1 (Remarks to the Author):

Review of NCOMMS-16-20310

Comments for transmission to the authors, and for the editor:

- Who will be interested in reading the paper, and why?

I think the paper will be of interest to structural geologists, geophysicists, geodynamicists and petrologists – although not all will understand the details of the model presented.

- What are the main claims of the paper and how significant are they?

The authors provide convincing evidence of past presence of aqueous fluids – in the form of specific types of olivine lattice preferred orientation (LPO) and newly formed hydrous phases (amphibole) in mantle peridotite. These are heterogeneously distributed but systematically related to other microstructural parameters such as strength of LPO, grain size, phase arrangement, and geometric properties of grain boundaries, that are generally accepted to indicate different deformation mechanisms and (possibly) strain magnitudes. They present a carefully considered and novel model for the way the fluid distribution evolved as deformation was accommodated. I think it is a good model, but I also think that the way it is described is not optimally clear. I am concerned that only researchers who are familiar with microstructural mechanisms and evidence will fully grasp the significance as it is presently presented. That said – I think that some refinement of the discussion could yield a model that is more understandable and significantly enhance the impact of the paper. I have tried to make suggestions to facilitate this in writing on the PDF of the manuscript and supporting info.

- Is the paper likely to be one of the five most significant papers published in the discipline this year?

Among researchers familiar with the micromechanisms of lithospheric deformation – Yes. However, I think that the impact will be less outside this specific field unless the model can be presented more clearly.

- How does the paper stand out from others in its field?

The authors claim that no-one has previously presented 'coeval evidence of water draining and strain localisation' in lithospheric rocks. I guess this is true... previous publications that I immediately think of include Fusses et al. (2009) who provided evidence of a permeability structure that evolves with deformation but no confirmed evidence of fluids, and Menegon et al. (2014) who demonstrated fluids must have been present (and their infiltration induced by creep dilatancy) in ductile shear zones in lower crustal rocks due to the formation of hydrous mineral assemblages, but focussed on the shear zone itself – and different composition layers within it – rather than demonstrating variation in less or more strained parts. I note that the authors do not cite the latter publication and probably should.

- Are the claims novel? If not, which published papers compromise novelty?

Yes, I think they are novel – see previous comment.

- Are the claims convincing? If not, what further evidence is needed?

The claims are fairly convincing but I think two key datasets are not clearly presented – if they can be then this is worthy of publication in Nature. These are:

1. Information about how grain size distributions were determined.
2. Variation in strain accommodated within protomylonites vs mylonites vs ultra mylonites is inferred but no offset marker or other measure of strain is presented here. This is a major downfall

Additionally, I would like to see better documentation of what processing/cleaning was carried out to the EBSD data. Finally I have a minor concern that the number of grains represented in the LPOs is very much at the lower limit of the number required to robustly determine the texture strength indices they use = J and M index. I did similar analyses on peridotites a few years ago, and found that the M indices I was calculating were quite dependant on the number of grains included. They could remove this source of uncertainty by redoing get the calculations illustrated in Fig. 3 for exactly the same number of randomly selected grains for each sample = 148.

- Are there other experiments or work that would strengthen the paper further?

I think they have to prove that more strain was accommodated in the finest layers – I think it would be robust to show an offset marker in other layers with comparable grain size distributions in the massif – you don't have to show it for these particular layers.

- How much would further work improve it, and how difficult would this be? Would it take a long time?

I suspect that provision of the additional information I suggest should be able to be accomplished easily – within days if the authors have time to devote to this process and if they have already documented a deformed marker on a similar shear zone that demonstrates that the strain accommodated in their different layers is indeed different. If these data don't yet exist a bit more fieldwork may be required and that would probably take longer.

- Are the claims appropriately discussed in the context of previous literature?

A discussion of Menegon et al. (2014) should be included. Otherwise, it is comprehensive.

- If the manuscript is unacceptable, is the study sufficiently promising to encourage the authors to resubmit?

Yes, definitely.

- If the manuscript is unacceptable but promising, what specific work is needed to make it acceptable?

Address the points I have made above, refine the writing style, and address my numerous hand-written annotations on the m.s.

Other questions to consider

- Is the manuscript clearly written?

The writing style is a little convoluted. I often felt the same point could have been made in half as many words. I have made numerous suggestions that I think will fix that in annotations on the attached PDF. I apologise for not writing these into a word document cross-referenced to line no and I hope the authors are able to translate the comments to that form in preparing a response.

- Would readers outside the discipline benefit from a schematic of the main result to accompany publication?

Figure 5 presents this quite well already. I think some simple graphs of shear strain across the various layers would make it even better.

- Could the manuscript be shortened? (Because of pressure on space in our printed pages we aim to publish manuscripts as short as is consistent with a persuasive message.)

The current writing could be reduced in length through more careful writing, but additional information is needed so probably not.

- Should the authors be asked to provide supplementary methods or data to accompany the paper online? (Such data might include source code for modelling studies, detailed experimental protocols or mathematical derivations.)

Grain size data, info on EBSD data processing/cleaning should be included in the supplementary info.

- Have the authors done themselves justice without overselling their claims?

Yes. However, as noted on annotations on the last para of your conclusion, I think you can apply this model to the shallower lithosphere as well as the mantle and that doing so would enhance the impact of the paper.

- Is the statistical analysis of the data sound, and does it conform to the journal's guidelines?

Grain size data and info on EBSD data processing/cleaning are needed.

References:

Fliervoet, T.F., White, S.H., and Drury, M.R., 1997, Evidence for dominant grain boundary sliding deformation in greenschist- and amphibolite-grade polymineralic ultramylonites from the Redbank Deformed Zone, Central Australia: *Journal of Structural Geology*, v. 19, p. 1495–1520, doi:10.1016/S0191-8141(97)00076-X

Fusseis, F., Regenauer-Lieb, K., Liu, J., Hough, R.M., and De Carlo, F. (2009) Creep cavitation can

establish a dynamic granular fluid pump in ductile shear zones. *Nature* 459: 974-977, doi: 10.1038/nature08051.

Hiraga, T., Miyazaki, T., Yoshida, H., Zimmerman, M.E., 2013. Comparison of microstructures in superplastically deformed synthetic materials and natural mylonites: Mineral aggregation via grain boundary sliding. *Geology* 41(9), 959-962, doi: 10.1130/G34407.1.

Menegon, L., Fusses, F., Stünitz, H., Xiao, X., 2015. Creep cavitation bands control porosity and fluid flow in lower crustal shear zones. *Geology*, 43, 227-230, doi:10.1130/G36307.1

Reviewer #2 (Remarks to the Author):

This is a very interesting study that reports detailed microstructural and crystallographic preferred orientation analysis of olivine in mantle shear zones from the Ronda mylonitic complex, southern Spain. The results are interpreted in terms of the relationship between the grain-scale deformation mechanisms of olivine and water distribution in the shear zone, and the Authors argue that progressive grain size reduction results in water draining to fill the new grain boundaries.

The Authors' main overarching conclusion is that ductile flow exerts a dynamic control on water-rich fluid distribution in mantle shear zones. This is certainly plausible, but in my view the data presented in this paper do not allow to fully support the model proposed. For this reason, the manuscript should be rejected.

This study is very interesting and timely, and in my view it can be significantly strengthened by the presentation of additional data and by a more comprehensive discussion. I hope that the Authors will find my comments useful to prepare a new version of the manuscript for a future submission.

1) The occurrence of E-type vs A-type (wet vs dry) CPO could also result from the presence of originally more hydrated vs dry olivine grains, and does not have to result necessarily from a dynamic fluid-pumping. For example, it has been shown that mantle shear zones can preferentially initiate within more hydrated domains of peridotite, which eventually developed an E-type CPO (e.g. Skemer et al. *EPSL* 2013).

Admittedly, whether the more hydrated conditions in the ultramylonites is a primary feature or the results of a dynamic evolution of the microstructure is difficult to resolve, but the Authors should mention that there are alternative scenarios. Some questions arise: why should olivine dry during deformation? What is the driving force for the loss of intracrystalline water from olivine grains with an E-type CPO to fill newly formed grain boundaries in the fine-grained ultramylonite? This model is intriguing, but at this stage it looks still very speculative to me. As the Authors conclude that there is a switch in olivine slip system (and of water content) with decreasing distance from the ultramylonite, a more systematic study of the protolith (and of strain gradients in general) is required. The CPO of the protolith is apparently a mix of E-type and A-type (lines 118-122) and this also could suggest an original heterogeneous distribution of aqueous fluids.

2) It strikes me that there is no systematic measurement of intracrystalline water contents in olivine and pyroxenes (with FTIR or SIMS) from the different microstructural domains of the shear zone. Also, no mass-balance calculation to estimate the amount of fluid infiltration in different domains was attempted. Dry vs wet conditions were concluded only on the basis of the olivine CPO, but more data is needed to clearly prove them.

3) If amphibole is a synkinematic phase, the Authors could possibly use Ti-in-amphibole to better constrain the P, T conditions during water-assisted shearing in the ultramylonite, to expand the Supplementary Figure 3. This would be an important information, which could potentially better constrain the conditions at which fluids were channelized in the fine-grained ultramylonite as

attested by the growth of pargasite in dilatant sites and quadruple junctions.

4) As a follow-up comment, it would be nice to quantify the enrichment in pargasite in the ultramylonite (mentioned for example in figure 2) and to identify the pargasite-forming reaction to try to place some constraints on the amount of water that was possibly channelized through the shear zone.

5) EBSD cannot distinguish between tilt and twist boundaries, it only identifies generic low angle boundaries that we interpret as tilt or twist boundaries on the basis of specific crystallographic relationships. The crystallographic data presented in this paper are consistent with tilt boundary models, but the systematic use of tilt boundaries to refer to generic low angle boundaries detected by EBSD is misleading and should be avoided.

6) From lines 109-111 and from the supplementary material, I was under the impression that the shear zone has been active over a broad range of P, T conditions. This most likely resulted in evolving microstructures and stress conditions as well. Which specific P, T range is the estimated differential stress (100-200 MPa) representative of (line 183)?

7) Please define what do you mean by "quasi-perfect phase mixing" (line 106): a random phase distribution? An anticorrelated phase distribution?

Reviewer #3 (Remarks to the Author):

Review of Précigout et al - Water pumping by grain size reduction during upper mantle flow

The paper explores deformation microstructures from rocks samples collected in the Ronda massif, Spain. It describes microfabrics reflecting various stages of mylonitic overprint and identifies mechanisms that likely dominated deformation in the various samples. Strain localisation is characterised by a transition from dislocation creep in mylonitic portions to grain size sensitive creep in ultramylonitic strands of the shear zone, where deformation presumably involved a component of viscous grain boundary sliding.

The authors use EBSD data to derive a conceptual model for the hydration state of the shear zone where the fine-grained ultramylonites attract water by providing excess surface area, and thereby dry/drain nearby mylonites, enforcing a change in the slip systems accommodating dislocation creep there. A critical aspect of their interpretation is a link between crystallographic orientation data from olivine domains with an inferred water content, which builds in previous work by Karato and others.

I find this a quite stimulating paper. It is well-written, shows very interesting data and it constructively contests a model for synkinematic permeability that I formulated in 2009. In trying to follow their arguments, a few questions arose though, which I listed below.

1) How does the model outlined by the authors advance the ideas presented in a range of papers by Bruce Watson's group from the early 2000s, most notably the model outlined in Wark and Watson (2000)?

Watson 1999 - American Mineralogist, 84, 1693–1710

Wark & Watson 2000 - GEOPHYSICAL RESEARCH LETTERS, 27/14, 2029-2032

Nakamura & Watson 2001 - Geofluids 1, 73–89

2) I am somewhat confused concerning the spatiotemporal sequence of events. Would drying of mylonitic olivines (i.e. the opposite of hydrolytic weakening) not drastically strengthen these domains, terminating any further deformation there, and partitioning all of the deformation into the ultramylonites?

3) The "dynamic long term fluid pumping" seems to be associated to the establishment of the fine-grained ultramylonites (cf. Wark and Watson, 2000). How do the authors envisage the

hydraulic gradients in the rock to be maintained once that happened and an ultramylonitic steady-state fabric has been established? Would then not VGBS initiate a granular fluid pump in the sense of my 2009 paper?

4) How was GSS creep and VGBS initiated in the first place, i.e. why did strain localize? I find this important step relatively poorly supported by the presented observations.

5) Which space did the enstatite nucleate in if not creep cavities, and what was the role of pressure solution/reprecipitation during deformation of the ultramylonite?

6) Have the authors considered supporting the inferred water contents in their samples with FTIR measurements? As I understand it, their interpretation hinges on the EBSD data, which give only indirect evidence of a lattice-bound water content. Actual measurements would massively strengthen this paper.

7) Could the authors show a high-resolution BSE image of the ultramylonitic microfabrics or, alternatively, a microtomographic dataset?

8) Lastly, how did the authors arrive at the conclusion that phase mixing is quasi-perfect?

Point-to-point answer to the reviewers
(The reviewer comments are in italic; our answers are in regular bold)

Reviewer #1:

Review of NCOMMS-16-20310

Comments for transmission to the authors, and for the editor:

- *Who will be interested in reading the paper, and why?*

I think the paper will be of interest to structural geologists, geophysicists, geodynamicists and petrologists – although not all will understand the details of the model presented.

- *What are the main claims of the paper and how significant are they?*

The authors provide convincing evidence of past presence of aqueous fluids – in the form of specific types of olivine lattice preferred orientation (LPO) and newly formed hydrous phases (amphibole) in mantle peridotite. These are heterogeneously distributed but systematically related to other microstructural parameters such as strength of LPO, grain size, phase arrangement, and geometric properties of grain boundaries, that are generally accepted to indicate different deformation mechanisms and (possibly) strain magnitudes. They present a carefully considered and novel model for the way the fluid distribution evolved as deformation was accommodated. I think it is a good model, but I also think that the way it is described is not optimally clear. I am concerned that only researchers who are familiar with microstructural mechanisms and evidence will fully grasp the significance as it is presently presented. That said – I think that some refinement of the discussion could yield a model that is more understandable and significantly enhance the impact of the paper. I have tried to make suggestions to facilitate this in writing on the PDF of the manuscript and supporting info.

According to these points, we substantially improved the manuscript by adding further petrological (pseudosection), structural (finite strain) and textural (EBSD) data (new figures 2, 3 and 4). They all confirm what we proposed in the first version, except with the fact that we give more credits to creep cavitation as a source for water pumping. We also simplified the title and clarified the discussion by developing our argumentation and by improving the last figure (new figure 8). Our modifications led us to add a new figure (new figure 2) and split the old figures 3 and 5, giving rise to 8 figures in total.

- *Is the paper likely to be one of the five most significant papers published in the discipline this year?*

Among researchers familiar with the micromechanisms of lithospheric deformation – Yes. However, I think that the impact will be less outside this specific field unless the model can be presented more clearly.

Please, see the comments above.

- *How does the paper stand out from others in its field?*

The authors claim that no-one has previously presented ‘coeval evidence of water draining and strain localisation’ in lithospheric rocks. I guess this is true... previous publications that I immediately think of include Fuisseis et al. (2009) who provided evidence of a permeability structure that evolves with deformation but no confirmed evidence of fluids, and Menegon et al. (2014) who demonstrated fluids must have been present (and their infiltration induced by creep dilatancy) in ductile shear zones in lower crustal rocks due to the formation of hydrous mineral assemblages, but focused on the shear zone itself – and different composition layers within it - rather than demonstrating variation in less or more strained parts. I note that the authors do not cite the latter publication and probably should.

We would like to insist that no one study gave evidence of dynamic water draining in ductile shear zone so far, particularly in the mantle. Based on observations of micro-cavities (Fusseis et al., 2009), dissolution-precipitation and nucleation of nominally hydrous phases in ductile shear bands (Menegon et al., 2015), the authors deduced that creep cavitation may result in water pumping to fill the strain-induced cavities. But they do not provide evidence that water filling occurred during plastic flow and strain localization. Our documentations of olivine fabric in Ronda demonstrate that ductile shear zone are pumping source for water. Furthermore, because water gradients are preserved into the mineral fabric, this dynamic draining necessarily occurs on a long-term basis. We discuss this point from line 233 to line 247. We also added the reference “Menegon et al., 2015” in the paper.

• *Are the claims novel? If not, which published papers compromise novelty?*

Yes, I think they are novel – see previous comment.

• *Are the claims convincing? If not, what further evidence is needed?*

The claims are fairly convincing but I think two key datasets are not clearly presented – if they can be then this is worthy of publication in Nature. These are:

1. Information about how grain size distributions were determined.

For the protolith and core of protomylonites, we used the intercept-length method. Elsewhere, i.e., the rim of protomylonites, mylonites and ultramylonites, we used EBSD map considering grain boundaries as defined by correlated misorientation angles higher than 10°. We added these points in methods section (line 332).

2. Variation in strain accommodated within protomylonites vs mylonites vs ultra mylonites is inferred but no offset marker or other measure of strain is presented here. This is a major downfall.

About this point, a bar graph has been added in the new figure 4. This graph documents the shear strain (γ) based on the angle between the long axis of olivine grains and shear plane, such as deduced from ultramylonitic layers (Ramsay, 1980).

Additionally, I would like to see better documentation of what processing/cleaning was carried out to the EBSD data.

We added a paragraph in the methods section (lines 325-335) to better describe the procedure of post-acquiring treatment of EBSD data, particularly for EBSD maps.

Finally I have a minor concern that the number of grains represented in the LPOs is very much at the lower limit of the number required to robustly determine the texture strength indices they use = J and M index. I did similar analyses on peridotites a few years ago, and found that the M indices I was calculating were quite dependant on the number of grains included. They could remove this source of uncertainty by redoing get the calculations illustrated in Fig. 3 for exactly the same number of randomly selected grains for each sample = 148.

In the new version, we substantially improved the new figure 4 by adding EBSD data. According to Skemer et al. (2005), the minimum amount of grains to calculate an objective M index is 150. All fabrics that we show in the new figure 4 are above this value and most of them are even above 180 grains. Furthermore, we better characterized the olivine LPO across the Ronda shear zones through calculation of the BA index (Mainprice et al., 2014), which gives information on the fabric geometry between a point maximum LPO (E-type fabric) and axial-type LPO (D-type fabric). Our new documentation further supports multi-scale, syn-tectonic water pumping towards ultramylonites.

• *Are there other experiments or work that would strengthen the paper further?*

I think they have to prove that more strain was accommodated in the finest layers – I think it would be robust to show an offset marker in other layers with comparable grain size distributions in the massif – you don't have to show it for these particular layers.

As shown in the new figure 4B, the angle between the long axis of olivine grains and

plane of ultramylonitic layers is very low ($< 10^\circ$), lower than everywhere else in the Ronda shear zones. This value indicates a finite shear strain higher than $\gamma = 5$, and hence, high-strain deformation (Ramsay, 1980).

- *How much would further work improve it, and how difficult would this be? Would it take a long time?*

I suspect that provision of the additional information I suggest should be able to be accomplished easily – within days if the authors have time to devote to this process and if they have already documented a deformed marker on a similar shear zone that demonstrates that the strain accommodated in their different layers is indeed different. If these data don't yet exist a bit more fieldwork may be required and that would probably take longer.

- *Are the claims appropriately discussed in the context of previous literature?*

A discussion of Menegon et al. (2015) should be included. Otherwise, it is comprehensive.

We added and discussed the study from Menegon et al. (2015) in the new version of the manuscript.

- *If the manuscript is unacceptable, is the study sufficiently promising to encourage the authors to resubmit?*

Yes, definitely.

- *If the manuscript is unacceptable but promising, what specific work is needed to make it acceptable?*

Address the points I have made above, refine the writing style, and address my numerous hand-written annotations on the m.s.

All points have been addressed.

Other questions to consider

- *Is the manuscript clearly written?*

The writing style is a little convoluted. I often felt the same point could have been made in half as many words. I have made numerous suggestions that I think will fix that in annotations on the attached PDF. I apologise for not writing these into a word document cross-referenced to line no and I hope the authors are able to translate the comments to that form in preparing a response.

All suggestions have been taken into account in the new version.

- *Would readers outside the discipline benefit from a schematic of the main result to accompany publication?*

Figure 5 presents this quite well already. I think some simple graphs of shear strain across the various layers would make it even better.

The last figure (now figure 8) has been improved to show how strain distributes across the Ronda shear zones, and not only around ultramylonites.

- *Could the manuscript be shortened? (Because of pressure on space in our printed pages we aim to publish manuscripts as short as is consistent with a persuasive message.)*

The current writing could be reduced in length through more careful writing, but additional information is needed so probably not.

We really tried to make the paper as short as possible. The current version is 518 words longer than the previous version.

- *Should the authors be asked to provide supplementary methods or data to accompany the paper online? (Such data might include source code for modelling studies, detailed experimental protocols or mathematical derivations.)*

Grain size data, info on EBSD data processing/cleaning should be included in the supplementary info.

Information about grain size calculation and EBSD data treatment has been added in the methods section.

• *Have the authors done themselves justice without overselling their claims?*

Yes. However, as noted on annotations on the last para of your conclusion, I think you can apply this model to the shallower lithosphere as well as the mantle and that doing so would enhance the impact of the paper.

In the last paragraph, we strongly suggest that our findings and model may be also valid in crustal rocks.

• *Is the statistical analysis of the data sound, and does it conform to the journal's guidelines?*

Grain size data and info on EBSD data processing/cleaning are needed.

They have been added in the method sections.

Reviewer #2 (Remarks to the Author):

This is a very interesting study that reports detailed microstructural and crystallographic preferred orientation analysis of olivine in mantle shear zones from the Ronda mylonitic complex, southern Spain. The results are interpreted in terms of the relationship between the grain-scale deformation mechanisms of olivine and water distribution in the shear zone, and the Authors argue that progressive grain size reduction results in water draining to fill the new grain boundaries.

The Authors' main overarching conclusion is that ductile flow exerts a dynamic control on water-rich fluid distribution in mantle shear zones. This is certainly plausible, but in my view the data presented in this paper do not allow to fully support the model proposed. For this reason, the manuscript should be rejected.

This study is very interesting and timely, and in my view it can be significantly strengthened by the presentation of additional data and by a more comprehensive discussion. I hope that the Authors will find my comments useful to prepare a new version of the manuscript for a future submission.

As said before, we substantially improved the manuscript through 1) adding further data (pseudosections, structural data and EBSD data) and 2) clarifying the introduction and discussion. Our new dataset further supports what we proposed before, but we give more credits to fluid pumping induced by creep cavitation. We also propose that subsequent phase nucleation may enhance long-term fluid pumping.

1) The occurrence of E-type vs A-type (wet vs dry) CPO could also result from the presence of originally more hydrated vs dry olivine grains, and does not have to result necessarily from a dynamic fluid-pumping. For example, it has been shown that mantle shear zones can preferentially initiate within more hydrated domains of peridotite, which eventually developed an E-type CPO (e.g. Skemer et al. EPSL 2013). Admittedly, whether the more hydrated conditions in the ultramylonites is a primary feature or the results of a dynamic evolution of the microstructure is difficult to resolve, but the Authors should mention that there are alternative scenarios. Some questions arise: why should olivine dry during deformation? What is the driving force for the loss of intracrystalline water from olivine grains with an E-type CPO to fill newly formed grain boundaries in the fine-grained ultramylonite? This model is intriguing, but at this stage it looks still very speculative to me. As the Authors conclude that there is a switch in olivine slip system (and of water content) with decreasing distance from the ultramylonite, a more systematic study of the protolith (and of strain gradients in general) is required. The CPO of the protolith is apparently a mix of E-type and A-type (lines 118-122) and this also could suggest an original heterogeneous distribution of aqueous fluids.

In the new version, we added more EBSD data that better describe the olivine fabric in the study area. They show that olivine LPOs are all typical of E-type fabric, but some of them combine with D-type fabric. We confirmed and quantified this combination using the

BA index (Mainprice et al., 2014). Moreover, we show that the fabric distribution correlates with the distribution of finite strain across mylonitic complexes, with water accumulation around ultramylonites. These points definitely support “dynamic” draining during strain localization instead of “passive” infiltration, particularly considering the occurrence of water gradient preserved in the mineral fabric at a millimeter scale. In this case, we can expect olivine to be dried where water is no longer available at grain boundaries. We discuss these features from line 233 to line 279.

2) *It strikes me that there is no systematic measurement of intracrystalline water contents in olivine and pyroxenes (with FTIR or SIMS) from the different microstructural domains of the shear zone. Also, no mass-balance calculation to estimate the amount of fluid infiltration in different domains was attempted. Dry vs wet conditions were concluded only on the basis of the olivine CPO, but more data is needed to clearly prove them.*

To better characterize the amount of water involved, we performed X calculations on modal compositions in ultramylonites. We focused on an amphibole-rich area for which we estimated around 600 ppm H_2O . We may expect lower water content, but the absence of plagioclase implies a minimum of 400 ppm (see new figure 3 and supplementary figure 3). We also chose to not perform FTIR analyses for several reasons, including the rate of water diffusion (Demouchy and Mackwell, 2003) and the fact that the Ronda peridotites lie on the crust since more than 20 Ma (Précigout et al., 2013). We justified our choice in the part “in-situ water content” of the methods section (lines 337-353).

3) *If amphibole is a synkinematic phase, the Authors could possibly use Ti-in-amphibole to better constrain the P, T conditions during water-assisted shearing in the ultramylonite, to expand the Supplementary Figure 3. This would be a important information, which could potentially better constrain the conditions at which fluids were channelized in the fine-grained ultramylonite as attested by the growth of pargasite in dilatant sites and quadruple junctions.*

We chose to not apply Ti-in-amphibole thermobarometers because the available ones (Ernst and Liu, 1998, American Mineralogist) have been calibrated for mafic rocks that commonly contain rutile. Because this latter is saturated in Titanium, it guaranties that Ca-amphiboles are in equilibrium with P-T conditions. In ultramafic rocks, we have no phase saturated in Ti. So, amphiboles are not necessarily in equilibrium with P-T conditions.

4) *As a follow-up comment, it would be nice to quantify the enrichment in pargasite in the ultramylonite (mentioned for example in figure 2) and to identify the pargasite-forming reaction to try to place some constraints on the amount of water that was possibly channelized through the shear zone.*

In the new version (Fig.3E), we provide a pseudosection that gives an estimation of water content in ultramylonites. We also give in supplementary material two pseudosections that constrains the minimum water content in ultramylonites.

5) *EBSDBSD cannot distinguish between tilt and twist boundaries, it only identifies generic low angle boundaries that we interpret as tilt or twist boundaries on the basis of specific crystallographic relationships. The crystallographic data presented in this paper are consistent with tilt boundary models, but the systematic use of tilt boundaries to refer to generic low angle boundaries detected by EBSDBSD is misleading and should be avoided.*

We better clarified this point in the new version of the manuscript. We also specify (line 190-192) that sub-grain boundaries are commonly assumed to be mostly tilt boundaries, so the main slip system can be deduced from them, as shown experimentally (Hansen et al., 2012).

6) *From lines 109-111 and from the supplementary material, I was under the impression that the shear zone has been active over a broad range of P, T conditions. This most likely resulted in evolving microstructures and stress conditions as well. Which specific P, T range is the estimated*

differential stress (100-200 MPa) representative of (line 183)?

We agree that deformation occurred over a wide range of pressures and temperatures in Ronda. But estimating P-T conditions from the olivine piezometer is extremely hazardous, even almost impossible in a context of strain localization. Indeed, stress strongly relies on strain rate (Hirth and Kohlstedt, 2003). For a given temperature and pressure, stress will highly change depending on the degree of weakening and strain localization.

7) Please define what do you mean by “quasi-perfect phase mixing” (line 106): a random phase distribution? An anticorrelated phase distribution?

We wanted to say that phases of the same nature are almost at equal distance from each other. This term has been deleted in the new version for clarity.

Reviewer #3 (Remarks to the Author):

Review of Précigout et al - Water pumping by grain size reduction during upper mantle flow

The paper explores deformation microstructures from rocks samples collected in the Ronda massif, Spain. It describes microfabrics reflecting various stages of mylonitic overprint and identifies mechanisms that likely dominated deformation in the various samples. Strain localisation is characterised by a transition from dislocation creep in mylonitic portions to grain size sensitive creep in ultramylonitic strands of the shear zone, where deformation presumably involved a component of viscous grain boundary sliding.

The authors use EBSD data to derive a conceptual model for the hydration state of the shear zone where the fine-grained ultramylonites attract water by providing excess surface area, and thereby dry/drain nearby mylonites, enforcing a change in the slip systems accommodating dislocation creep there. A critical aspect of their interpretation is a link between crystallographic orientation data from olivine domains with inferred water content, which builds in previous work by Karato and others.

I find this a quite stimulating paper. It is well-written, shows very interesting data and it constructively contests a model for synkinematic permeability that I formulated in 2009. In trying to follow their arguments, a few questions arose though, which I listed below.

1) How does the model outlined by the authors advance the ideas presented in a range of papers by Bruce Watson's group from the early 2000s, most notably the model outlined in Wark and Watson (2000)?

Watson 1999 - American Mineralogist, 84, 1693–1710

Wark & Watson 2000 - GEOPHYSICAL RESEARCH LETTERS, 27/14, 2029-2032

Nakamura & Watson 2001 - Geofluids 1, 73–89

In the new version, the studies of the Watson's group have been considered, which led to modify the introduction and discussion. Furthermore, the new version of our paper gives more credits to water pumping induced by creep cavitation in mantle shear zone. We further propose that phase nucleation may enhance long-term water pumping. We also definitely show that the distribution of olivine fabric in Ronda cannot be an effect of “passive” draining induced by grain size reduction.

2) I am somewhat confused concerning the spatiotemporal sequence of events. Would drying of mylonitic olivines (i.e. the opposite of hydrolytic weakening) not drastically strengthen these domains, terminating any further deformation there, and partitioning all of the deformation into the ultramylonites?

Based on a recent study (Fei et al., 2013), the degree of weakening induced by

hydrolytic weakening is quite low with respect to the one induced by grain size reduction in the diffusion creep regime (the one that occurs in ultramylonites). Nevertheless, the deformation is fully ductile – except for cavitation –, so the deformation cannot fully stop at one place while it goes on just by; the strain rate only decreases.

3) *The "dynamic long term fluid pumping" seems to be associated to the establishment of the fine-grained ultramylonites (cf. Wark and Watson, 2000). How do the authors envisage the hydraulic gradients in the rock to be maintained once that happened and an ultramylonitic steady-state fabric has been established? Would then not VGBS initiate a granular fluid pump in the sense of my 2009 paper?*

We fully agree that the fabric distribution in Ronda, particularly at a millimeter-scale, requires a “dynamic” pumping. We changed the paper accordingly.

4) *How was GSS creep and VGBS initiated in the first place, i.e. why did strain localize? I find this important step relatively poorly supported by the presented observations.*

As suggested by Hidas et al. (2016), we propose in the new version that dissolution-precipitation at pyroxene boundaries has been involved at the very beginning to trigger strain localization. So the ultramylonites were the first layers produced, and then the mylonites, and protomylonites result from subsequent water pumping towards ultramylonites. Very new experimental results support this feature (Précigout and Stünitz, 2016). We synthesize this point in the new figure 8 (figure 5 in the last version). We also propose that creep cavitation has enhanced dissolution-precipitation towards ultramylonites.

5) *Which space did the enstatite nucleate in if not creep cavities, and what was the role of pressure solution/reprecipitation during deformation of the ultramylonite?*

Please, see the comments above.

6) *Have the authors considered supporting the inferred water contents in their samples with FTIR measurements? As I understand it, their interpretation hinges on the EBSD data, which give only indirect evidence of a lattice-bound water content. Actual measurements would massively strengthen this paper.*

As mentioned above to answer to reviewer #2, we chose to not perform FTIR analyses for several reasons, including the rate of water diffusion (Demouchy and Mackwell, 2003) and the fact that the Ronda peridotites lie on the crust since more than 20 Ma (Précigout et al., 2013). We justified our choice in the “in-situ water content” part of the methods section.

7) *Could the authors show a high-resolution BSE image of the ultramylonitic microfabrics or, alternatively, a microtomographic dataset?*

We added a BSE image of an ultramylonitic layer in the new figure 3. We also tried to perform some microtomography in the past, but the expected cavities are below the resolution limit. Nevertheless, we identify very few micro-pores on the BSE image (~ 1 μm).

8) *Lastly, how did the authors arrive at the conclusion that phase mixing is quasi-perfect?*

We wanted to say that phases of the same nature are almost at equal distance from each other. However, this term has been deleted in the new version for clarity.

REVIEWERS' COMMENTS:

Reviewer #1 (Remarks to the Author):

The additional information you have provided address all my questions about the work. Thank you. I am now recommending publication, subject to some minor editing of english. Please look carefully at lines:

99 (suggests not suggest)

120 (ultramylonites needs an apostrophe)

191-192

206

244 (more or less not of less)

265-266

275-276

343-346

352

and revise the English in company of / on advice of a native speaker if possible.

Also, you say that you have addressed all my hand-written annotations about revisions to writing style, but when I look at the new abstract it is nearly identical to the previous one, despite my numerous annotations recommending revision. You should respond to this.

Reviewer #2 (Remarks to the Author):

Dear Authors, dear Editor,

The revised manuscript has thoroughly addressed most of the reviewers' main comments. All the additional data and figures are necessary to strengthen the paper and to clarify aspects of the methods and results. The proposed model is much more convincing now, and in my view the paper in the revised form represents a high-impact contribution that highlights the importance of creep cavitation and of fluid-rock interactions for mantle dynamics.

I have only a few very minor comments that I invite the Authors to consider.

1. In the rebuttal letter the Authors insist that no one study gave evidence of dynamic water draining in ductile shear zones so far, and in the introduction (line 68) they state that enrichment in amphibole together with changes in dislocation slip-system in olivine document syn-tectonic water draining in the Ronda shear zones. Please note that Menegon et al. (2015) also documented a syn-tectonic fluid draining in ultramylonites deforming by GSS creep (though in lower crustal shear zones) on the basis of chemical analysis showing an increase in aqueous and carbonic fluids (resulting in syn-tectonic growth of amphibole and calcite) in lower crustal ultramylonites with respect to their protolith. The sentence in lines 65-67 should probably be slightly modified to consider these results as well.
2. I understand the answer to my original comment 6). But this is exactly the reason why an estimate of differential stress from olivine grain size is very difficult in these rocks due to the prolonged deformation history under varying P, T (and presumably stress-strain rate?) conditions. In my view the differential stress of 100-200 MPa reported in line 220 is not a critical information for this paper and could be omitted.
3. Line 177: the J-index given here does not correspond to the one shown in Fig. 5B, B3 area.
4. Line 260. I agree with reviewer 3 that progressive drying of olivine could lead to hardening, and the Authors' answer has clarified this point. In the discussion of strain partitioning the Authors could mention the results of Fei et al. (2013) and highlight that the weakening due to grain size reduction/phase mixing is expected to outweigh the one induced by hydrolytic weakening.

Point-by-point response to the reviewers

Reviewer #1

The additional information you have provided address all my questions about the work. Thank you. I am now recommending publication, subject to some minor editing of english. Please look carefully at lines:

99 (*suggests not suggest*)

244 (*more or less not of less*)

These two mistakes have been corrected.

120 (*ultramylonites needs an apostrophe*)

Based on the comments of our native-english colleague (Prof. Jessica Warren), this correction is apparently not relevant.

191-192

206

265-266

275-276

343-346

352

The reviewer #1 did not leave any comment for these lines. Prof. Jessica Warren (see below) and we double-checked all of those, but they do not seem to be incorrect.

and revise the English in company of / on advice of a native speaker if possible.

Prof. Jessica Warren accepted to read the final version of the manuscript. As a specialist of rock deformation and a native-English researcher, she looked carefully at the English writing. The corrections are included in the “tracked” version of the manuscript.

Also, you say that you have addressed all my hand-written annotations about revisions to writing style, but when I look at the new abstract it is nearly identical to the previous one, despite my numerous annotations recommending revision. You should respond to this.

Although we were quite surprised by this comment because we took into account most of the reviewer’s comments for the abstract (as shown in the pdf file of our last submission), we have improved it (now 148 words instead of 155) and we modified its last sentence to insist on the implications of our findings.

Reviewer #2:

Dear Authors, dear Editor,

The revised manuscript has thoroughly addressed most of the reviewers’ main comments. All the additional data and figures are necessary to strengthen the paper and to clarify aspects of the methods and results. The proposed model is much more convincing now, and in my view the paper in the revised form represents a high-impact contribution that highlights the importance of creep cavitation and of fluid-rock interactions for mantle dynamics.

I have only a few very minor comments that I invite the Authors to consider.

1. *In the rebuttal letter the Authors insist that no one study gave evidence of dynamic water*

draining in ductile shear zones so far, and in the introduction (line 68) they state that enrichment in amphibole together with changes in dislocation slip-system in olivine document syn-tectonic water draining in the Ronda shear zones. Please note that Menegon et al. (2015) also documented a syn-tectonic fluid draining in ultramylonites deforming by GSS creep (though in lower crustal shear zones) on the basis of chemical analysis showing an increase in aqueous and carbonic fluids (resulting in syn-tectonic growth of amphibole and calcite) in lower crustal ultramylonites with respect to their protolith. The sentence in lines 65-67 should probably be slightly modified to consider these results as well.

In the new version, we better considered the study of Menegon et al. (2015) and changed the sentence in lines 65-67 accordingly.

2. I understand the answer to my original comment 6). But this is exactly the reason why an estimate of differential stress from olivine grain size is very difficult in these rocks due to the prolonged deformation history under varying P, T (and presumably stress-strain rate?) conditions. In my view the differential stress of 100-200 MPa reported in line 220 is not a critical information for this paper and could be omitted.

We definitely agree with reviewer 2 about this point, but we would like to keep this estimation for one specific reason: E-type fabric has been only documented below 300 MPa of differential stress (Karato et al., 2001). At higher stresses, we would expect B-type fabric to occur. Although approximative, our estimations give an idea of the amount of stress we can expect during deformation in Ronda.

3. Line 177: the J-index given here does not correspond to the one shown in Fig. 5B, B3 area.

The figure has been corrected in the new version

4. Line 260. I agree with reviewer 3 that progressive drying of olivine could lead to hardening, and the Authors' answer has clarified this point. In the discussion of strain partitioning the Authors could mention the results of Fei et al. (2013) and highlight that the weakening due to grain size reduction/phase mixing is expected to outweigh the one induced by hydrolytic weakening.

About this point, we added a sentence at lines 261-263, which includes a reference to the study of Fei et al. (2013).